# How Well Do Low-PRAL Diets Fare in Comparison to the 2020–2025 Dietary Guidelines for Americans?

**DOI:** 10.3390/healthcare11020180

**Published:** 2023-01-06

**Authors:** Maximilian Andreas Storz, Alvaro Luis Ronco

**Affiliations:** 1Center for Complementary Medicine, Department of Internal Medicine II, Freiburg University Hospital, Faculty of Medicine, University of Freiburg, 79106 Freiburg, Germany; 2Unit of Oncology and Radiotherapy, Pereira Rossell Women’s Hospital, Bvard. Artigas 1590, Montevideo 11600, Uruguay

**Keywords:** alkaline diet, potential renal acid load, low-PRAL diet, macronutrients, minerals, vitamins, deficiencies, ash–alkaline diet

## Abstract

The regular consumption of net acid-producing diets can produce “acid stress” detrimental to human health. Alkalizing diets characterized by a negative potential renal acid load (also called low-PRAL diets (LPD)) enjoy uninterrupted popularity. However, the nutritional adequacy of said diets has rarely been assessed in large populations. Using data from the National Health and Nutrition Examination Surveys, we estimated nutrient intake in individuals consuming an LPD and contrasted the results in an age- and sex-specific manner to individuals on an acidifying diet (high-PRAL diet, HPD). Both groups were compared with the daily nutritional goals (DNG) specified in the 2020–2025 Dietary Guidelines for Americans (DGA). Our analysis included 29,683 individuals, including 7234 participants on an LPD and 22,449 participants on an HPD. Individuals on an LPD numerically met more nutritional goals than individuals on an HPD, yet both failed to meet the goals for several nutrients of public health concern (vitamin D and calcium). As opposed to individuals on an HPD, LPD consumers met the DGA recommendations for saturated fat and potassium. Individuals on an LPD consumed significantly more fiber than individuals on an HPD, as well as yielded a more favorable potassium-to-sodium intake ratio.

## 1. Introduction

Consumption of a net acid-producing diet can produce “acid stress” detrimental to human health [1,2]. Progressive acid accumulation subsequent to a high intake of acid precursors and impaired buffer mechanisms can alter the systemic acid–base balance [3,4]. Chronic low-grade acid accumulation has been associated with cardiometabolic diseases [5,6,7], renal disorders [8,9,10], and various cancer types [11,12,13,14].

There is now compelling evidence that a diet rich in fruits and vegetables beneficially affects dietary acid load (DAL) [15,16]. DAL is determined by the net balance of acidifying foods and alkalizing foods [16]. One of the most commonly employed DAL estimates is potential renal acid load (PRAL) [17]. The PRAL value is an established method to estimate the potential acid load of foods (usually in mEq/100g edible food) [17]. The PRAL concept is physiologically based, and it was developed more than two decades ago by Remer and Manz [17,18]. Positive overall PRAL scores (>0 mEq/day) indicate a net acidifying diet, whereas negative overall PRAL scores (<0 mEq/day) indicate an alkalizing diet.

Said alkalizing diets (also called low-PRAL diets (LPD) or alkali–ash diets) enjoy uninterrupted popularity [19]. Adherence to LPD has been associated with several health benefits, including an improved potassium-to-sodium ratio, reduced muscle wasting, and increased intracellular magnesium concentrations [19]. Compared to LPD, diets with a high acid load may be detrimental to human health—particularly in the elderly [20]. A Korean study in middle-aged and older Korean adults found a positive association between a higher DAL and the incidence of hyperuricemia [21]. A study by Li et al. revealed a positive association between a higher DAL and the risk of hip fractures in an elderly Chinese population [22]. Kataya et al. reported positive associations with the prevalence of frailty (particularly slowness/weakness and low physical activity) in older Japanese women [23].

LPD are characterized by a high intake of alkalizing foods (vegetables, fruits, legumes, various nuts, and seeds) and limited amounts of meat and dairy products, which are abundant in acid precursors (mainly sulfur-containing amino acids (cysteine, homocysteine, and methionine) and preservative phosphate)) [24,25,26].

Although several studies emphasized that alkaline diets may exert beneficial effects toward human health [27,28], some studies also reported that excess diet alkalinity may be detrimental [29,30]. Notably, the nutritional adequacy of alkalizing diets has rarely been assessed in large populations. Whether alkaline diets are in line with current dietary guidelines is virtually unknown.

Using nationally representative cross-sectional data for the US general population, we estimated nutrient intake in individuals consuming an alkalizing diet (an LPD with a PRAL-sum <0 mEq/day) and contrasted the results in an age- and sex-specific manner to individuals on an acidifying diet (PRAL-sum >0 mEq/day). We also contrasted both groups to the daily nutritional goals (DNGs) specified in the current 2020–2025 Dietary Guidelines for Americans (DGA) [31]. The main aim was to investigate for which nutrients the DNGs were met, and for which nutrients an insufficient (or excessive) intake occurred.

## 2. Materials and Methods

### 2.1. The NHANES

The NHANES is a cross-sectional and nationally representative US-based program of studies designed to assess the health and nutritional status of noninstitutionalized citizens in the United States [32,33]. NHANES is conducted by the National Center for Health Statistics, which is part of the Centers for Disease Control and Prevention. The survey encompasses approximately 5000 participants per annum, and it includes demographic, dietary, socioeconomic, and health-related interview questions. All interviews are standardized and conducted in participants’ homes. Health measurements and instrumental diagnostic procedures are performed in specially equipped and designed mobile examination centers. A detailed description of the NHANES may be obtained from the official NHANES homepage (https://www.cdc.gov/nchs/nhanes/index.htm) (accessed on 11 December 2022) [33].

### 2.2. Study Population and DAL Assessment

For the current study, we merged and appended multiple NHANES modules, including the dietary interview module and the demographics public release file [34,35]. All estimates of energy and nutrient intakes were obtained from the first day of the dietary recall. Information on all nutrients, minerals, and vitamins displayed in the DNGs (Table A1-2) in the current DGA were extracted (page 131) [31]. Atwater’s values for the metabolizable energy of macronutrients were used to calculate the percentage of total energy from each macronutrient [36,37].

An alkaline diet (LPD) was defined by a total PRAL-value <0 mEq/day (negative PRAL value). An acidifying diet (high-PRAL diet (HPD) was defined by a total PRAL-value ≥0 mEq/day (positive PRAL value).

PRAL was calculated using the following formula [17]:PRAL (mEq/day) = (0.49 × total protein (g/day)) + (0.037 × phosphorus (mg/day)) − (0.021 × potassium (mg/day)) − (0.026 × magnesium (mg/day)) − (0.013 × calcium (mg/day)).(1)

This formula by Remer et al. takes into account ionic dissociation, intestinal absorption rates for the included micro- and macronutrients and sulfur metabolism [17,18]. Previous studies in healthy individuals demonstrated a moderate-to-strong correlation between the PRAL score and urinary pH.

We excluded all participants with an implausible energy intake (either ≤800 kcal/day or ≥5000 kcal/day) and implausible PRAL-values (<−100 mEq/day or >100 mEq/day). No data imputation took place, and participants with missing data on any nutrient relevant for the current were excluded. Nutrient intake profiles of both groups were then descriptively compared to the DNGs in the current DGA.

### 2.3. Dietary Guidelines for Americans

The DGA aim to provide food-based recommendations to promote health and to help prevent diet-related diseases [31,38]. Published by the US Department of Agriculture and the US Department of Health and Human Services, the DGA are a central element of US Federal nutrition policy and nutrition education activities. The current version of the DGA has 164 pages and intends to guide nutrition and health professionals to support all individuals consume a healthy, nutritionally adequate diet. A special section is dedicated to dietary components of public health concern for underconsumption (page 36 pp) [31]. Specific age- and sex-based recommendations can be found in the appendix (page 131 pp) [31,38]. DNGs are available for females and males, stratified by specific age groups. These include (I) individuals aged 19–30 years, (II) individuals aged 31–50 years, and (III) individuals aged 51 years or older. For the present analysis, we used this classification and descriptively compared nutrient intake in adult NHANES participants on an LPD/HPs with the DNGs stratified by age-sex groups.

As discussed in detail in one of our previous publications [39], the DNGs in the current DGA stem from various sources. In brief, concepts and sources include adequate intake (AI), acceptable macronutrient distribution range (AMDR), chronic disease reduction level (CDRR), dietary guidelines for Americans (DGA), and the recommended dietary allowance (RDA).

The present analysis considers all nutrients, vitamins, and minerals included in the DNGs Table A1-2 of the current DGA (2020–2025) [31]. Nutrients included the three macronutrients (reported in g/day and in %/total kcal intake), saturated fat, fiber, and two polyunsaturated fatty acids (linoleic acid and linolenic acid). Vitamins included vitamin A, B1, B2, B3, B6, B12, C, E, K, and folate. Minerals encompassed calcium, iron, magnesium, phosphorus, potassium, sodium, and zinc.

### 2.4. Statistics

The NHANES sample is selected through a complex, multistage probability design [32]. We used the appropriate Stata survey commands for all statistical procedures in order to account for population weights and the complex. All statistical analyses were conducted in Stata software version 14 (StataCorp., College Stadion, TX, USA). We performed unconditional subclass analyses (preserving the main survey design and providing larger standard errors) to estimate nutrient intake in individuals on an LPD. Six NHANES survey cycles were appended (2007–2008, 2009–2010, 2011–2012, 2013–2014, 2015–2016, and 2017–2018) to increase the total sample size.

We described normally distributed variables with their mean and standard error in parentheses. Categorical variables were described with the number of observations (n) and with weighted proportions (with their corresponding standard error) in parentheses.

To allow for a more convenient comparison between estimated nutrient intakes in participants on an LPD/HPD and the DNGs in the DGA, we used color coding for all tables. A red-colored box indicates a violation of the DGA recommendations. As indicated by the arrow direction, this could be either an insufficient intake (shown by an arrow pointing downward) or an excessive intake (shown by an arrow pointing upward). Green color indicates that the observed intake was in accordance with the DNG. For energy intake, we used orange color coding.

We performed all comparisons in an entirely descriptive way without statistical testing with the exception of a final comparison of nutrient intake the entire sample across PRAL-quartiles. For this, we used Stata’s design-adjusted Wald test to examine potential differences across the four PRAL quartiles. All tests were two-sided, and statistical significance was determined at α = 0.05.

## 3. Results

### 3.1. Sample Characteristics

The total study sample comprised n = 29,683 individuals, including n = 7234 participants on an LPD and n = 22,449 participants on an HPD. Table 1 shows the sample characteristics in detail. The weighted proportion of females on an LPD was significantly higher as compared to males (59.45% vs. 40.55%). Participants’ ethnicity is also shown in Table 1. Of note, significant intergroup differences in the weighted proportions were observed for non-Hispanic Whites, Mexican Americans, and non-Hispanic Blacks.

The mean PRAL of individuals on an LPD was −12.67 (0.23) mEq/day, indicating an alkalizing potential. The mean PRAL of individuals on an HPD was 23.35 (0.20) mEq/day, indicating acidifying properties.

### 3.2. Macronutrient and Fiber Intake

Macronutrient and fiber intake of participants on an LPD in comparison to the DNGs in the current DGA is displayed in Table 2 (males) and Table 3 (females).

Both sexes met the macronutrient goals (AMDR) in the current DGA. Moreover, both sexes met the DGA recommendations for saturated fat intake. As expected, males on an LPD had a higher total energy intake as compared to females. The data also suggest that both sexes failed to meet the daily nutritional goals for fiber.

In a similar pattern, Table 4 and Table 5 display macronutrient and fiber intake in males (Table 4) and females (Table 5) on an HPD.

Unlike participants on an LPD, males aged 51 years or older did not meet the macronutrient goals (AMDR) for total fat intake. A similar pattern was observed for females across all age groups. Additionally, said individuals did not meet the DGA recommendations for saturated fatty acid intake (Table 4 and Table 5). Fiber intake in participants on an HPD was substantially lower as compared to individuals on an LPD, particularly in males.

### 3.3. Mineral and Vitamin Intake

Table 6, Table 7, Table 8 and Table 9 show mineral and vitamin intake in participants on an LPD (Table 6 and Table 7) and on an HPD (Table 8 and Table 9).

Males on an LPD (Table 6) met the recommendations for iron, phosphorus, and potassium. They also met the recommendations for calcium, except in the age group 51+ years. This pattern was also observed with regard to zinc. Potassium intake exceeded the 3400 mg margin in all age groups. An insufficient intake was observed for vitamins A, E, and D, as well as choline (Table 6).

Notably, we observed a different picture in females on an LPD (Table 7). None of the female age groups met the DNGs for calcium and magnesium. DNGs for iron were only met in females aged 51 years or older. In comparison to males on an LPD, females aged 31 years or older met the DNGs for vitamin A. Similar to males, females did not meet the DNGs for vitamins E and D, as well as choline.

In a similar style, Table 8 and Table 9 show mineral and vitamin intake in participants on an HPD.

When compared to individuals on an LPD, there were several noticeable differences. Neither males (Table 8) nor females (Table 9) met the DNGs for potassium. Sodium intake was substantially higher as compared to individuals on an LPD. A comparable picture was observed for vitamins A, E, D, and C, as well as magnesium and choline. Of note, males on an HPD aged 19 to 50 years did not meet the DNGs for vitamin K.

Lastly, we examined nutrient intake in the entire sample stratified by PRAL-quartiles (Table 10). Quartile intervals are shown in Table 10. Light-blue coloring was used for descending trend across quartiles, whereas orange color indicates an ascending trend across quartiles. Total lipid intake and saturated fatty acid intake significantly increased across quartiles (*p* < 0.001), whereas carbohydrate and vitamin C intake significantly decreased across quartiles (*p* < 0.001). As expected, potassium intake substantially decreased over quartiles Q1–Q3, whereas sodium and phosphorus intake increased across quartiles (*p* < 0.001). Additional significant trends were found for zinc, niacin, choline, and vitamin B12.

## 4. Discussion

The present study assessed nutrient intake in NHANES participants on an LPD in comparison to the daily nutritional goals specified in the current 2020–2025 DGA [31], and in comparison to individuals on an HPD. As opposed to individuals on an HPD, LPD consumers met the recommendations for (saturated) fat and potassium intake (both sexes). Although individuals on an LPD consumed more fiber than individuals on an HPD, they did not meet the daily nutrition goal for fiber in the DGA. Several noticeable and significant trends were observed across PRAL-quartiles, such as a lower sodium and phosphorus intake in individuals on an LPD.

Much has been written in the lay literature on the benefits of the alkaline diet (19), yet, to the best of our knowledge, its nutritional quality has not been assessed in large cross-sectional studies in comparison to current established dietary guidelines. This is important, because alkaline diets attract extensive attention—not only in health-minded healthy individuals but also in patients with chronic noncommunicable diseases.

One of many prominent examples is the prevention and treatment of cancer. Two independent meta-analyses revealed a positive association between a high DAL and various cancer types [40,41]. An elevated DAL has been associated with an increased total mortality and breast cancer-specific mortality in breast cancer survivors [42]. Elevated DAL scores were also significantly associated with inflammation, reduced overall physical health, and poorer glycemic control in said individuals [43,44]. In patients with advanced pancreatic cancer, alkalization therapy may enhance the effects of chemotherapy [45]. Despite the limited number of studies in this field [46], many individuals afflicted with cancer are interested in dietary modifications using alkaline diets.

Here, we assessed the nutritional quality of said LPD/alkaline diets. The mean PRAL of individuals on an LPD was negative (−12.67 (0.23)), indicating clearly alkalizing properties. Individuals on an LPD met more nutritional goals (numerically) than individuals on an HPD, particularly with regard to potassium and fat intake. In line with previous publications [47,48], individuals on an LPD demonstrated a more favorable potassium-to-sodium ratio. Notably, even individuals on an LPD exceeded the recommendations for sodium, and did not meet the DNGs for several nutrients of public health concern, including vitamin D and calcium (particularly in women).

The nutrient intake differences across PRAL-quartiles may be at least partly explained by total energy intake, which varied substantially between quartile 1 and quartile 4 (Table 1). This also results in a significantly higher protein intake in Q2–Q4 as compared to Q4. Protein has the highest weighting factor in the PRAL-formula and may contribute substantially to the PRAL differences across quartiles.

The approximately U-shaped calcium and magnesium intake patterns across quartiles may be surprising at a first glance. However, when adjusting for total intake (not shown), intake for both micronutrients was lower in quartile 4 than in quartile 1. As such, the results are completely in line with the expectations toward an LPD (as opposed to an HPD).

Despite the cross-sectional nature of our data, we conclude that an alkalizing diet is not necessarily healthful and in full accordance with all nutritional goals specified in the DGA. Attention has to be paid with regard to nutrients of public health concern, and supplementation might be warranted—particularly with regard to magnesium.

Strengths of the current analysis include the modest sample size and the nationally representative data set from the National Health and Nutrition Examination Survey. To the best of our knowledge, we are the first group to systematically assess the nutritional quality of LPD in comparison to the current DGA. The large sample allowed for analyses stratified by age–sex groups as found in the 2020–2025 DGA. Weaknesses included the sparse study population characterization and the cross-sectional nature of our data, which did not allow for causal interferences. The group assignment to either an LPD or an HPD (based solely on the PRAL value and the cutoff of 0 mEq/day) is potentially controversial and dogmatic. On the other hand, further breaking down the sample in, for example, a “very alkaline diet group” and a “moderately alkaline group” could introduce bias, as reference ranges for PRAL values are nonexistent and subject to debate. Lastly, one must take into account that the PRAL formula by Remer and Manz does not adjust for energy intake. Diets with a higher energy intake which do not restrict protein would, thus, inevitably result in higher PRAL scores (see Table 10). Although of potential interest, we did not calculate PRAL scores based on energy-adjusted nutrient intakes since the PRAL formula by Remer and Manz has not been validated with regard to this aspect.

## 5. Conclusions

Alkalizing diets enjoy uninterrupted popularity—despite limited clinical evidence derived from randomized-controlled trials. Several studies reported potential health benefits associated with alkaline diets. In line with previous reports, our data suggest that an LPD is associated with a more favorable potassium-to-sodium ratio in the NHANES. Individuals on an LPD met more nutritional goals than individuals on an HPD, particularly with regard to (saturated) fat intake. Yet, caution is warranted due to the cross-sectional nature of our data and its inherent limitations. Our results suggest that meeting the DNG for several nutrients of public health concern (particularly vitamin D and calcium) appears to be difficult on an LPD. In light of the growing clinical importance of alkalizing diets, additional trials investigating their nutritional adequacy are urgently warranted in this area. Investigations in other large cohorts and in special populations (e.g., pregnant women and patients afflicted with cancer) are also required for a better understanding of the nutritional adequacy of LPD. Once said data are available, specific supplementation strategies could be developed for individuals choosing LPD.

## Figures and Tables

**Table 1 healthcare-11-00180-t001:** Sample characteristics. The sample included n = 7234 participants on an LPD and n = 22,449 participants on an HPD.

	Individuals on an LPD			Individuals on an HPD		
	Number of Observations (n)	Weighted Proportion (%)	±SE	Number of Observations (n)	Weighted Proportion (%)	±SE
**Age**						
19–30 years	n = 985	15.15	0.80	n = 5009	24.29	0.67
31–50 years	n = 2030	30.97	0.97	n = 7709	36.30	0.65
51 years and older	n = 4219	53.88	1.21	n = 9731	39.41	0.70
**Sex**						
Male	n = 3033	40.55	0.73	n = 11,487	50.78	0.43
Female	n = 4201	59.45	0.73	n = 10,962	49.22	0.43
**Race/Ethnicity**						
Mexican American	n = 898	6.74	0.64	n = 3637	9.42	0.83
Other Hispanic	n = 844	5.86	0.53	n = 2236	5.78	0.46
Non-Hispanic White	n = 3274	69.44	1.55	n = 9096	65.30	1.42
Non-Hispanic Black	n = 1291	9.24	0.69	n = 5002	11.75	0.78
Other Race—including multiracial	n = 927	8.72	0.57	n = 2478	7.76	0.45

**Table 2 healthcare-11-00180-t002:** Macronutrient and fiber intake in males following an LPD compared to the DNGs in the 2020–2025 DGA stratified by age group.

Macronutrients and Fiber	Source of Goal	LPD M 19–30		DGA M 19–30	DNG Met?	LPD M 31–50		DGA M 31–50	DNG Met?	LPD M 51+		DGA M 51+	DNG Met?
		Mean	±SE			Mean	±SE			Mean	±SE		
Energy intake (kcal/day)		2355.71	78.43	2400	↓	2355.34	49.19	2200	↑	2079.86	28.11	2000	↑
Protein (% kcal)	AMDR	12.29	0.24	10–35		12.92	0.22	10–35		13.69	0.16	10–35	
Protein (g)	RDA	71.34	2.46	56	↑	74.39	1.83	56	↑	69.31	0.80	56	↑
Carbohydrate (% kcal)	AMDR	56.31	0.91	45–65		54	0.58	45–65		52.54	0.46	45–65	
Carbohydrate (g)	RDA	327.39	11.82	130	↑	314.20	6.02	130	↑	270.28	3.55	130	↑
Fiber (g)	14 g/1000 kcal	22.67	0.88	34	↓	23.73	0.73	31	↓	21.53	0.46	28	↓
Total lipid (% kcal)	AMDR	30.28	0.49	20–35		30.28	0.49	20–35		32.08	0.44	20–35	
Saturated fatty acids (% kcal)	DGA	8.78	0.25	<10		9.25	0.19	<10		10.05	0.15	<10	↑
18:2 linoleic acid (g)	AI	17.33	0.91	17		17.51	0.71	17		16.30	0.49	14	
18:3 linolenic acid (g)	AI	1.63	0.09	1.6		1.81	0.09	1.6		1.84	0.06	1.6	

AMDR = acceptable macronutrient distribution range, RDA = recommended dietary allowance, AI = adequate intake. Green-colored boxes indicate a nutrient intake in accordance with the DNG, whereas red-colored boxes indicate a violation of the DNG recommendations.

**Table 3 healthcare-11-00180-t003:** Macronutrient and fiber intake in females following an LPD compared to the DNGs in the 2020–2025 DGA stratified by age group.

Macronutrients and Fiber	Source of Goal	LPD F 19–30		DGA F 19–30	DNG Met?	LPD F 31–50		DGA F 31–50	DNG Met?	LPD F 51+		DGA F 51+	DNG Met?
		Mean	±SE			Mean	±SE			Mean	±SE		
Energy intake (kcal/day)		1767.41	33.70	2000	↓	1763.64	26	1800	↓	1621.48	13.57	1600	↑
Protein (% kcal)	AMDR	12.36	0.27	10–35		12.97	0.16	10–35		13.76	0.13	10–35	
Protein (g)	RDA	53.08	1.25	46	↑	55.89	0.91	46	↑	54.76	0.65	46	↑
Carbohydrate (% kcal)	AMDR	55.16	0.67	45–65		55.34	0.42	45–65		53.63	0.38	45–65	
Carbohydrate (g)	RDA	241.85	5.07	130	↑	242.53	3.81	130	↑	216.02	2.36	130	↑
Fiber (g)	14 g/1000 kcal	17.55	0.51	28	↓	19.40	0.53	25	↓	18.48	0.28	22	↓
Total lipid (% kcal)	AMDR	31.39	0.33	20–35		31.39	0.33	20–35		32.25	0.29	20–35	
Saturated fatty acids (% kcal)	DGA	9.68	0.21	<10		9.63	0.13	<10		10.09	0.12	<10	↑
18:2 linoleic acid (g)	AI	13.65	0.43	12		13.78	0.32	12		12.81	0.19	11	
18:3 linolenic acid (g)	AI	1.45	0.05	1.1		1.58	0.05	1.1		1.47	0.03	1.1	

AMDR = acceptable macronutrient distribution range, RDA = recommended dietary allowance, AI = adequate intake. Green-colored boxes indicate a nutrient intake in accordance with the DNG, whereas red-colored boxes indicate a violation of the DNG recommendations.

**Table 4 healthcare-11-00180-t004:** Macronutrient and fiber intake in males following an HPD compared to the DNG in the 2020–2025 DGA stratified by age group.

Macronutrients and Fiber	Source of Goal	HPD M 19–30		DGA M 19–30	DNG Met?	HPD M 31–50		DGA M 31–50	DNG Met?	HPD M 51+		DGA M 51+	DNG Met?
		Mean	±SE			Mean	±SE			Mean	±SE		
Energy intake (kcal/day)		2560.34	25.30	2400	↑	2615.78	21.57	2200	↑	2337.45	18.35	2000	↑
Protein (% kcal)	AMDR	16.42	0.17	10–35		16.65	0.12	10–35		16.73	0.12	10–35	
Protein (g)	RDA	102.39	1.39	56	↑	105.77	0.98	56	↑	95.33	0.80	56	↑
Carbohydrate (% kcal)	AMDR	46.79	0.30	45–65		45.39	0.23	45–65		44.82	0.22	45–65	
Carbohydrate (g)	RDA	298.36	3.64	130	↑	295.91	3.01	130	↑	261.17	2.39	130	↑
Fiber (g)	14 g/1000 kcal	16.82	0.32	34	↓	18.13	0.27	31	↓	17.63	0.26	28	↓
Total lipid (% kcal)	AMDR	33.90	0.2	20–35		34.73	0.21	20–35		36.10	0.18	20–35	↑
Saturated fatty acids (% kcal)	DGA	11.27	0.10	<10	↑	11.41	0.09	<10	↑	11.76	0.08	<10	↑
18:2 linoleic acid (g)	AI	19.09	0.33	17		19.86	0.25	17		18.83	0.26	14	
18:3 linolenic acid (g)	AI	1.93	0.04	1.6		1.98	0.03	1.6		1.96	0.04	1.6	

AMDR = acceptable macronutrient distribution range, RDA: recommended dietary allowance, AI = adequate intake. Green-colored boxes indicate a nutrient intake in accordance with the DNG, whereas red-colored boxes indicate a violation of the DNG recommendations.

**Table 5 healthcare-11-00180-t005:** Macronutrient and fiber intake in females following an HPD compared to the DNG in the 2020–2025 DGA stratified by age group.

Macronutrients and Fiber	Source of Goal	HPD F 19–30		DGA F 19–30	DNG Met?	HPD F 31–50		DGA F 31–50	DNG Met?	HPD F 51+		DGA F 51+	DNG Met?
		Mean	±SE			Mean	±SE			Mean	±SE		
Energy intake (kcal/day)		2011.11	17.31	2000	↑	1988.65	14.57	1800	↑	1823.75	16.12	1600	↑
Protein (% kcal)	AMDR	15.72	0.13	10–35		16.31	0.10	10–35		16.72	0.12	10–35	
Protein (g)	RDA	76.85	0.76	46	↑	78.58	0.63	46	↑	73.91	0.71	46	↑
Carbohydrate (% kcal)	AMDR	48.26	0.29	45–65		47.14	0.28	45–65		46.08	0.30	45–65	
Carbohydrate (g)	RDA	242.45	2.59	130	↑	232.96	2.08	130	↑	209.91	2.32	130	↑
Fiber (g)	14 g/1000 kcal	14.02	0.24	28	↓	15.07	0.21	25	↓	14.78	0.20	22	↓
Total lipid (% kcal)	AMDR	35.01	0.24	20–35	↑	35.12	0.20	20–35	↑	36.59	0.23	20–35	↑
Saturated fatty acids (% kcal)	DGA	11.65	0.10	<10	↑	11.58	0.10	<10	↑	11.92	0.10	<10	↑
18:2 linoleic acid (g)	AI	15.98	0.25	12		16.06	0.20	12		15.49	0.20	11	
18:3 linolenic acid (g)	AI	1.64	0.03	1.1		1.62	0.02	1.1		1.67	0.03	1.1	

AMDR = acceptable macronutrient distribution range, RDA: recommended dietary allowance, AI = adequate intake. Green-colored boxes indicate a nutrient intake in accordance with the DNG, whereas red-colored boxes indicate a violation of the DNG recommendations.

**Table 6 healthcare-11-00180-t006:** Mineral and vitamin intake in males following an LPD compared to the DNG in the 2020–2025 DGA stratified by age group.

Minerals and Vitamins	Unit	Source of Goal	LPD M 19–30		DGA M 19–30	DNG	LPD M 31–50		DGA M 31–50	DNG	LPD M 51+		DGA M 51+	DNG
			Mean	±SE			Mean	±SE			Mean	±SE		
Calcium	mg	RDA	1077.38	51.44	1000		1027.21	26.66	1000		970.98	16.90	1000	↓
Iron	mg	RDA	16.54	0.75	8		16.42	0.31	8		15.89	0.28	8	
Magnesium	mg	RDA	382.94	14.88	400	↓	390.79	9.18	420	↓	351.71	5.71	420	↓
Phosphorus	mg	RDA	1326.33	46.81	700		1363.79	31.59	700		1276.85	16.71	700	
Potassium	mg	AI	3468.22	105.92	3400		3651.62	69.98	3400		3465.28	44.06	3400	
Sodium	mg	CDRR	3566.41	136.48	2300	↑	3580.35	94.59	2300	↑	3224.19	44.61	2300	↑
Zinc	mg	RDA	11.09	0.47	11		11.12	0.27	11		10.81	0.18	11	↓
Vitamin A	mcg RAEd	RDA	624.025	46.90	900	↓	790.18	44.03	900	↓	754.47	20.95	900	↓
Vitamin E	mg ATd	RDA	11.07	0.87	15	↓	10.89	0.54	15	↓	9.44	0.24	15	↓
Vitamin D	IUDd	RDA	4.63 mcg	0.41	600	↓	4.53 mcg	0.27	600	↓	4.77 mcg	0.15	600	↓
Vitamin C	mg	RDA	174.53	13.98	90		148.83	6.75	90		131.69	5.16	90	
Thiamin	mg	RDA	1.78	0.07	1.2		2.35	0.05	1.2		1.68	0.03	1.2	
Riboflavin	mg	RDA	2.19	0.10	1.3		2.74	0.22	1.3		2.38	0.04	1.3	
Niacin	mg	RDA	27.58	1.16	16		27.19	0.74	16		24.13	0.35	16	
Vitamin B-6	mg	RDA	2.65	0.15	1.3		2.59	0.08	1.3		2.28	0.05	1.7	
Vitamin B-12	mcg	RDA	4.61	0.31	2.4		4.68	0.24	2.4		4.52	0.14	2.4	
Choline	mg	AI	311.74	13.39	550	↓	321.61	6.33	550	↓	324.01	5.29	550	↓
Vitamin K	mcg	AI	138.04	14.49	120		143.20	8.57	120		146.51	6.54	120	
Folate	mcg DFEd	RDA	651.39	31.63	400		623.41	19.85	400		591.74	12.11	400	

CDRR = chronic disease reduction level, RDA: recommended dietary allowance, AI = adequate intake. Green-colored boxes indicate a nutrient intake in accordance with the DNG, whereas red-colored boxes indicate a violation of the DNG recommendations.

**Table 7 healthcare-11-00180-t007:** Mineral and vitamin intake in females following an LPD compared to the DNG in the 2020–2025 DGA stratified by age group.

Minerals and Vitamins	Unit	Source of Goal	LPD F 19–30		DGA F 19–30	DNG	LPD F 31–50		DGA F 31–50	DNG	LPD F 51+		DGA F 51+	DNG
			Mean	±SE			Mean	±SE			Mean	±SE		
Calcium	mg	RDA	825.10	51.44	1000	↓	859.66	21.31	1000	↓	841.93	12.98	1200	↓
Iron	mg	RDA	11.95	0.35	18	↓	12.97	0.31	18	↓	12.67	0.18	8	
Magnesium	mg	RDA	283.18	6.57	310	↓	309.479	6.84	320	↓	292.15	3.13	320	↓
Phosphorus	mg	RDA	974.26	22.22	700		1057.48	18.17	700		1032.93	12.54	700	
Potassium	mg	AI	2655.56	53.75	2600		2845.94	51.72	2600		2800.87	31.06	2600	
Sodium	mg	CDRR	2612.77	66.60	2300	↑	2656.61	51.25	2300	↑	2491.67	28.95	2300	↑
Zinc	mg	RDA	8.13	0.25	8		8.52	0.17	8		8.64	0.14	8	
Vitamin A	mcg RAEd	RDA	624.27	31.59	700	↓	723.57	44.73	700		704.22	17.27	700	
Vitamin E	mg ATd	RDA	8.51	0.36	15	↓	8.64	0.28	15	↓	8.326	0.20	15	↓
Vitamin D	IUDd	RDA	3.19 mcg	0.17	600	↓	3.49 mcg	0.13	600	↓	3.94 mcg	0.09	600	↓
Vitamin C	mg	RDA	130.32	6.98	75		115.98	4.17	75		77.98	7.34	75	
Thiamin	mg	RDA	1.32	0.04	1.1		1.37	0.03	1.1		1.35	0.02	1.1	
Riboflavin	mg	RDA	1.74	0.07	1.1		1.85	0.04	1.1		1.87	0.02	1.1	
Niacin	mg	RDA	19.85	0.61	14		18.94	0.33	14		18.45	0.25	14	
Vitamin B-6	mg	RDA	2.00	0.07	1.3		1.85	0.05	1.3		1.77	0.03	1.5	
Vitamin B-12	mcg	RDA	3.53	0.18	2.4		3.42	0.12	2.4		3.53	0.09	2.4	
Choline	mg	AI	228.92	7.78	425	↓	239.16	4.82	425	↓	242.79	3.48	425	↓
Vitamin K	mcg	AI	158.18	10.91	90		182.29	15.62	90		152.40	5.50	90	
Folate	mcg DFEd	RDA	469.68	16.68	400		490.07	13.12	400		473.61	7.96	400	

CDRR = chronic disease reduction level, RDA: recommended dietary allowance, AI = adequate intake. Green-colored boxes indicate a nutrient intake in accordance with the DNG, whereas red-colored boxes indicate a violation of the DNG recommendations.

**Table 8 healthcare-11-00180-t008:** Mineral and vitamin intake in males following an HPD compared to the DNG in the 2020–2025 DGA stratified by age group.

Minerals and Vitamins	Unit	Source of Goal	HPD M 19–30		DGA M 19–30	DNG	HPD M 31–50		DGA M 31–50	DNG	HPD M 51+		DGA M 51+	DNG
			Mean	±SE			Mean	±SE			Mean	±SE		
Calcium	mg	RDA	1164.73	18.83	1000		1115.43	16.39	1000		999.32	13.70	1000	↓
Iron	mg	RDA	16.88	0.25	8		17.09	0.22	8		16.51	0.19	8	
Magnesium	mg	RDA	319.97	5.01	400	↓	342.00	3.48	420	↓	324.22	3.50	420	↓
Phosphorus	mg	RDA	1659.74	20.23	700		1714.72	16.61	700		1550.13	13.74	700	
Potassium	mg	AI	2697.90	38.44	3400	↓	2949.18	27.09	3400	↓	2866.85	26.52	3400	↓
Sodium	mg	CDRR	4342.44	55.65	2300	↑	4416.37	44.20	2300	↑	3965.69	36.70	2300	↑
Zinc	mg	RDA	13.86	0.22	11		14.42	0.20	11		13.35	0.25	11	
Vitamin A	mcg RAEd	RDA	619.97	15.59	900	↓	637.46	12.94	900	↓	685.86	29.73	900	↓
Vitamin E	mg ATd	RDA	9.07	0.23	15	↓	9.66	0.17	15	↓	9.32	0.14	15	↓
Vitamin D	IUDd	RDA	4.99 mcg	0.16	600	↓	5.19 mcg	0.17	600	↓	5.47 mcg	0.15	600	↓
Vitamin C	mg	RDA	74.69	2.53	90	↓	71.05	1.82	90	↓	71.03	1.32	90	↓
Thiamin	mg	RDA	1.94	0.04	1.2		1.90	0.02	1.2		1.81	0.02	1.2	
Riboflavin	mg	RDA	2.44	0.05	1.3		2.61	0.05	1.3		2.40	0.04	1.3	
Niacin	mg	RDA	33.68	0.59	16		33.57	0.48	16		28.77	0.30	16	
Vitamin B-6	mg	RDA	2.66	0.07	1.3		2.67	0.06	1.3		2.25	0.03	1.7	
Vitamin B-12	mcg	RDA	6.41	0.16	2.4		6.62	0.16	2.4		6.14	0.29	2.4	
Choline	mg	AI	384.92	6.01	550	↓	423.34	4.73	550	↓	396.04	3.78	550	↓
Vitamin K	mcg	AI	95.83	3.81	120	↓	111.48	3.98	120	↓	111.53	3.01	120	↓
Folate	mcg DFEd	RDA	627.40	12.68	400		600.64	9.48	400		562.80	8.08	400	

CDRR = chronic disease reduction level, RDA: recommended dietary allowance, AI = adequate intake. Green-colored boxes indicate a nutrient intake in accordance with the DNG, whereas red-colored boxes indicate a violation of the DNG recommendations.

**Table 9 healthcare-11-00180-t009:** Mineral and vitamin intake in females following an HPD compared to the DNGs in the 2020–2025 DGA stratified by age group.

Minerals and Vitamins	Unit	Source of Goal	HPD F 19–30		DGA F 19–30	DNG	HPD F 31–50		DGA F 31–50	DNG	HPD F 51+		DGA F 51+	DNG
			Mean	±SE			Mean	±SE			Mean	±SE		
Calcium	mg	RDA	930.72	12.83	1000	↓	920.27	11.18	1000	↓	848.92	12.48	1200	↓
Iron	mg	RDA	13.48	0.18	18	↓	13.53	0.16	18	↓	13.05	0.18	8	
Magnesium	mg	RDA	253.63	3.43	310	↓	277.94	2.88	320	↓	262.34	2.93	320	↓
Phosphorus	mg	RDA	1287.76	12.82	700		1321.15	10.88	700		1242.87	13.47	700	
Potassium	mg	AI	2153.29	25.63	2600	↓	2286.60	19.75	2600	↓	2261.30	22.64	2600	↓
Sodium	mg	CDRR	3406.51	32.04	2300	↑	3347.80	28.40	2300	↑	3061.10	27.59	2300	↑
Zinc	mg	RDA	10.37	0.11	8		10.55	0.11	8		9.94	0.12	8	↑
Vitamin A	mcg RAEd	RDA	547.44	13.24	700	↓	575.17	11.13	700	↓	602.29	12.10	700	↓
Vitamin E	mg ATd	RDA	7.67	0.16	15	↓	8.23	0.16	15	↓	7.92	0.15	15	↓
Vitamin D	IUDd	RDA	4.15 mcg	0.13	600	↓	4.29 mcg	0.11	600	↓	4.40 mcg	0.13	600	↓
Vitamin C	mg	RDA	64.18	1.94	75	↓	63.68	1.76	75	↓	62.21	1.21	75	↓
Thiamin	mg	RDA	1.50	0.02	1.1		1.46	0.02	1.1		1.39	0.02	1.1	
Riboflavin	mg	RDA	1.86	0.03	1.1		1.95	0.02	1.1		1.87	0.02	1.1	
Niacin	mg	RDA	23.77	0.33	14		23.29	0.21	14		21.34	0.26	14	
Vitamin B-6	mg	RDA	1.86	0.04	1.3		1.82	0.03	1.3		1.68	0.03	1.5	
Vitamin B-12	mcg	RDA	4.54	0.10	2.4		4.65	0.09	2.4		4.47	0.10	2.4	
Choline	mg	AI	289.88	4.24	425	↓	305.86	3.13	425	↓	295.77	3.56	425	↓
Vitamin K	mcg	AI	92.15	3.30	90		105.08	3.43	90		107.23	2.98	90	
Folate	mcg DFEd	RDA	502.696	10.23	400		479.12	6.85	400		454.13	6.66	400	

CDRR = chronic disease reduction level, RDA: recommended dietary allowance, AI = adequate intake. Green-colored boxes indicate a nutrient intake in accordance with the DNG, whereas red-colored boxes indicate a violation of the DNG recommendations.

**Table 10 healthcare-11-00180-t010:** Nutrient intake across PRAL-quartiles: an overview.

Nutrients	Q1 (PRAL: <0.36)		Q2 (PRAL: 0.36 to 13.33)		Q3 (PRAL: 13.34 to 27.30)		Q4 (PRAL: >27.30)		*p*-Value
	Mean	±SE	Mean	±SE	Mean	±SE	Mean	±SE	
PRAL (mEq/day)	−12.311	0.227	7.122	0.064	19.838	0.064	43.735	0.285	*p* < 0.001
Energy intake (kcal/day)	1898.08	13.70	1877.05	12.05	2109.15	11.49	2669.36	14.70	*p* < 0.001
Protein (% kcal)	13.317	0.076	14.818	0.073	16.077	0.079	18.602	0.111	*p* < 0.001
Carbohydrate (% kcal)	53.911	0.233	49.690	0.187	46.669	0.192	42.277	0.168	*p* < 0.001
Total lipid (% kcal)	31.635	0.199	33.920	0.152	35.248	0.144	36.947	0.150	*p* < 0.001
Fiber (g)	20.099	0.269	15.666	0.181	15.804	0.203	17.016	0.188	*p* < 0.001
Saturated fatty acids (% kcal)	9.836	0.078	10.924	0.077	11.582	0.064	12.346	0.079	*p* < 0.001
18:2 linoleic acid (g)	14.700	0.198	14.839	0.129	16.753	0.193	21.185	0.220	*p* < 0.001
18:3 linolenic acid (g)	1.624	0.027	1.548	0.018	1.724	0.023	2.143	0.026	*p* < 0.001
Calcium	908.651	8.942	849.205	9.362	935.993	9.013	1187.806	12.468	*p* < 0.001
Iron	14.102	0.154	13.198	0.116	14.393	0.118	17.697	0.136	*p* < 0.001
Magnesium	324.628	3.017	274.765	2.535	285.805	2.693	334.324	2.685	*p* < 0.001
Phosphorus	1145.510	9.546	1168.196	9.411	1359.695	8.643	1867.811	11.007	*p* < 0.001
Potassium	3077.719	22.629	2423.967	19.629	2415.221	19.378	2820.948	20.027	*p* < 0.001
Sodium	2897.185	25.780	3026.610	22.316	3491.650	22.475	4732.316	30.405	*p* < 0.001
Zinc	9.528	0.096	9.750	0.099	11.234	0.086	15.351	0.147	*p* < 0.001
Vitamin A	717.343	14.812	568.262	9.226	579.180	8.618	698.424	17.608	*p* < 0.001
Vitamin E	9.134	0.161	7.741	0.097	8.394	0.134	9.918	0.124	*p* < 0.001
Vitamin D	4.083	0.065	4.055	0.075	4.542	0.092	5.766	0.135	*p* < 0.001
Vitamin C	126.733	2.568	74.675	1.357	64.513	1.270	62.773	1.215	*p* < 0.001
Thiamin	1.501	0.014	1.439	0.013	1.578	0.013	1.979	0.018	*p* < 0.001
Riboflavin	2.047	0.020	1.898	0.018	2.080	0.020	2.613	0.030	*p* < 0.001
Niacin	21.589	0.208	21.813	0.182	25.267	0.219	34.820	0.310	*p* < 0.001
Vitamin B-6	2.071	0.023	1.824	0.024	1.988	0.024	2.624	0.039	*p* < 0.001
Vitamin B-12	3.950	0.064	4.263	0.085	4.989	0.065	7.229	0.188	*p* < 0.001
Choline	273.062	2.492	278.174	2.453	324.088	2.756	454.026	3.347	*p* < 0.001
Vitamin K	155.131	4.511	104.505	2.720	101.923	2.326	108.078	1.946	*p* < 0.001
Folate	531.103	6.861	483.626	5.738	506.063	4.851	614.846	6.499	*p* < 0.001

Light-blue color indicates a descending trend, whereas orange color indicates an ascending trend.

## Data Availability

Data are publicly available online (https://wwwn.cdc.gov/nchs/nhanes/Default.aspx; accessed on 2 July 2022). The datasets used and analyzed in the current study are available from the corresponding author upon reasonable request.

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
