# Peer review of "How Well Do Low-PRAL Diets Fare in Comparison to the 2020–2025 Dietary Guidelines for Americans?"

_healthcare, 2023, doi:10.3390/healthcare11020180_

Round 1

Reviewer 1 Report

Dear authors,

thank you for this very interesting paper.

I have only few remarks:

The discussion is very clear an delivers concrete answers on the questions of your investigation. in contrast to this, the conclusion is too short and general. "In light of the growing clinical importance of alkalizing diets, additional trials are urgently warranted in this area".

I suggest that you offer concrete necessary additional trials based on your results.

Author Response

Dear Reviewer,

We would like to thank you very much for careful and thorough reading of this manuscript and for the thoughtful and positive comments, which help us to improve the quality of this article. Please kindly find our response attached. All requested changes have been clearly marked in yellow and blue color.

Sincerely,

The authors

Reviewer 2 Report

The study is very interesting and focuses on the nutritional adequacy of alkalizing diets.

Authors should pay attention to the following points:

1) The introduction should be enriched with information about the importance of PRAL for people of different ages and gender.

2) Age groups should be indicated in materials and methods

3) Color coding should be written in notes to tables

4) Table 1 shows the distribution according to ethnicity. Does it have anything to do with research? This is not included in the materials and methods, as well as in the description of the results of the experiment.

5) When reporting PRAL quartiles in Table 10, intervals (perhaps percentiles) should be reported, not the mean. The indicated p value was obtained by comparing all quartiles? For quartile values 1 and 2 for 18:2 Linoleic acid and vitamin D p<0.001?

6) Will the PRAL quartiles for women and men be different?

7) Quite a large part of the discussion is devoted to diseases, which would be more appropriate when justifying the purpose of the study.

8) The discussion focuses on the discussion of the LPD diet. In table 10, when distributed by quartiles according to PRAL, only category Q1 refers to LPD (mean PRAL<0), the other three quartiles (mean PRAL>0), if I understood the description correctly, refer to HPD.

The discussion should be expanded to compare the quartiles according to their category (LPD and HPD), their difference is obvious, however, the authors do not comment on possible causes other than energy consumption. What is the reason for the increased energy consumption?

Author Response

(The authors gave the same response as above.)

Reviewer 3 Report

Dear authors,

In this study, nutrient intake in individuals consuming a LPD and contrasted the results in an age-sex specific manner to individuals on an acidifying diet (high-PRAL
diet, HPD). Both groups were compared with the daily nutritional goals (DNG) specified in the 2020–2025 Dietary Guidelines for Americans (DGA) was investigated. I think the novelty of the topic and the information included is quite interesting, however, the following improvements should be made in order to allow a better discussion for the results obtained.

-Line 39: add the abbreviation with Potential Renal Acid Load (PRAL), as mentioned for first time. Please check and apply this issue throughout the manuscript.

-Line 57: replace of (mEq/d) with (mEq/day).

-Line 74: write the website link of the official NHANES homepage and the accessed date.

-Line 87: write the reference number of “This formula by Remer et al.”

-Line 93 DGA, write the complete form first time.

In table 1, what is the mean of “Se” standard error? if yes, it should be ± SE, in addition, please define each symbol and abbreviation under the tables and figures.

-Line 156: move theses details under the table: AMDR = Acceptable Macronutrient Distribution Range, RDA: 1Recommended Dietary Allowance, AI = Adequate Intake. Also, write DGA DNG etc. as well. Also, write the meaning of each standard reference color (line arrow). Apply that for each table.

-In table 2, 3 categorize linoleic and linolenic acids as poly un saturated fatty acids (PUFAs).

Author Response

(The authors gave the same response as above.)
